# Agroecosystem edge effects on vegetation, soil properties, and the soil microbial community in the Canadian prairie

**Mariah Aguiar, Alexandra J. Conway, Jennifer K. Bell, Katherine J. Stewart** *

Department of Soil Science, University of Saskatchewan, Saskatoon, Saskatchewan, Canada

* Katherine.stewart@usask.ca

## Abstract

Edge effects resulting from adjacent land uses are poorly understood in agroecosystems yet understanding above and belowground edge effects is crucial for maintaining ecosystem function. The aim of our study was to examine impacts of land management on above-ground and belowground edge effects, measured by changes in plant community, soil properties, and soil microbial communities across agroecosystem edges. We measured plant composition and biomass, soil properties (total carbon, total nitrogen, pH, nitrate, and ammonium), and soil fungal and bacterial community composition across perennial grassland-annual cropland edges. Edge effects due to land management were detected both aboveground and belowground. The plant community at the edge was distinct from the adjacent land uses, where annual, non-native, plant species were abundant. Soil total nitrogen and carbon significantly decreased across the edge ($P < 0.001$), with the highest values in the perennial grasslands. Both bacterial and fungal communities were different across the edge with clear changes in fungal communities driven directly and indirectly by land management. A higher abundance of pathogens in the more heavily managed land uses (i.e. crop and edge) was detected. Changes in plant community composition, along with soil carbon and nitrogen also influenced the soil fungal community across these agroecosystems edges. Characterizing edge effects in agroecosystem, especially those associated with soil microbial communities, is an important first step in ensuring soil health and resilience in these managed landscapes.

## 1. Introduction

Habitat fragmentation is a leading cause of biodiversity loss [1, 2] and agriculture has caused extensive habitat fragmentation [3, 4]. Highly fragmented landscapes have a high proportion of edges, which affect various ecological aspects [5]. Edges can be high contrast such as a forest abutting a pasture, or a more gradual low contrast edge like a shrub patch adjacent a meadow. Edges have edge effects which are abiotic and biotic changes occurring at the bounds of an ecosystem or habitat patch [5, 6] influencing properties including microclimate, moisture, soils, plant or animal community composition and distribution [7–9]. Some factors that influence

**Data Availability Statement:** All soils and vegetation files are available from the Federated Research Data Repository (DOI: https://doi.org/10.20383/103.0623). Raw sequence fasta files and the associated metadata can be found at the National

Center for Biotechnology Information (NCBI) under Bioproject PRJNA588061.

**Funding:** This research was supported by a Natural Sciences and Engineering Research Council of Canada Strategic Partnership Grant entitled "Understanding resilience in agroecosystems: landscapes in transition" with Agriculture and Agri-Food Canada. The funders had no role in study design, data collection and analysis, decision to publish, or preparation of the manuscript.

**Competing interests:** The authors have declared that no competing interests exist.

edges are orientation [10, 11], time [12], patch size [13, 14], edge contrast [15] and matrix composition [16, 17]. Ecological dynamics and patterns around edges can be understood through four essential mechanisms, ecological flows across edges, resource distribution, resource mapping, and unique species interactions [6].

Expansion and intensification of agriculture has induced change in nearby habitats, and have been observed in both plant communities and soil properties [18–20]. Agricultural intensification is thought to magnify edge effects [19] further altering vegetation and soil biodiversity in these systems [21]. Commonly, edges in the agroecosystem are inhabited by non-native undesirable plants, here called weeds, or other invasive species [22]. Plant communities at the edge may be of concern to farmers, where weeds can compete with crops [23]. While aboveground vegetation changes at the edge are evident, belowground changes are also occurring. Underlying gradients of soil properties have been found at edges, including soil pH, nitrogen (N), and carbon (C) [24, 25], though these studies are limited to forest edges. Aboveground and belowground interactions are important to consider because those interactions determine ecosystem function, and in particular agroecosystems, where land management has effects beyond the field boundary. However, the extent and characteristics of edges and their effects in agroecosystems remain poorly understood belowground.

Two major land uses in the agroecosystem are cultivated croplands and grasslands; they each have characteristics that affect the soil microbial community. Nutrient dynamics between the two are quite different; for instance, croplands often have lower soil C than grasslands [26, 27] while grasslands have more soil C and are frequently correlated with higher microbial biomass [28]. Various environmental factors affect soil microbial community composition and function [29], but agricultural practices directly alter environmental conditions affecting soil microbes [30]. These agricultural practices include but are not limited to, soil amendments [31, 32], tillage [33, 34], herbicides [35, 36], and crop type [37]. However, the magnitude to which these factors influence the soil microbial community are complex [38–40]; considering edge effects and the interactions with agricultural practices is essential to understand soil microbial community dynamics in these landscapes.

Aboveground edge effects provide insight into belowground conditions and ultimately the soil microbial community. Plant species can have specific microbial associations affecting microbial community composition, such as mycorrhizal associations with plant roots [41]. Additionally, invasive plant species can alter the soil microbial community through changing inputs of litter quality and quantity [42]. Knowing how and what alters the soil microbial community is important, as soil microorganisms are critical in maintaining ecosystem function, especially through nutrient cycling, disease suppression, and plant growth promotion [43, 44].

Understanding how the soil microbial community responds to edge effects is crucial, as the soil microbial community is essential for maintaining ecosystem function, especially with intensification of agricultural lands [44]. To investigate edge effects in agroecosystems above and belowground, we measured vegetation composition and biomass, and soil physicochemical and microbial properties across perennial grassland and annual cropland edges in central Saskatchewan, Canada. Our goal was to determine if changes in land use altered the plant community and soil properties at agricultural edges, and if so, how these changes influenced the microbial community across the edge. Considering the interrelated effects of management on soil properties and plant communities, and their impacts on soil microbial communities, will better our understanding of agroecosystem edges and their ecosystem function.

## 2. Materials and methods

### 2.1. Study sites

We examined perennial grassland-annual cropland edges at two locations, St. Denis National Wildlife Area (SDNWA) and the Conservation Learning Centre (CLC), in southern-central Saskatchewan, Canada. SDNWA is located in the Moist Mixed Grassland ecoregion and CLC is in the Boreal Transition ecoregion [45]. Soils at SDNWA are mostly of Dark Brown Chernozemic and CLC are predominantly Black Chernozemic soils [46] and were confirmed by another study that was sampling cores for their study. Authorization to sample at these sites were granted by the St. Denis National Wildlife Area and the Conservation Learning Centre.

Both locations are composed of cropland interspersed with perennial grasslands. Both croplands are no-till, while perennial grasslands are not intensively managed, only being cut for hay and no grazing, fertilizing, or spraying occurs. At SDNWA, in 1977, 97 hectares of cropland were converted to a perennial forage predominately composed of smooth brome (*Bromus inermis* L.), alfalfa (*Medicago sativa* L.), and yellow sweet clover (*Melilotus officinale* L.) [47]. Perennial grasslands at both sites were cut once for hay in 2017 and croplands were planted with flax (*Linuum usitatissiumum* var. CDC Sorrel) in May 2017 at SDNWA. Glyphosate was applied prior to seeding and during seeding granular fertilizer (90 N—36 P– 17 S kg/hectare) was used; herbicides (Buctril M and Centurion mix) were also applied in July 2017 at SDNWA. Canola (*Brassica napus* L., Nexera RR112) was planted in May 2017 at CLC. At the time of seeding, anhydrous fertilizer was applied (112 N– 28 P– 28 S kg/hectare) as well as glyphosate. Fungicides were applied in June (Topnotch/Eclipse) and July (Lance) 2017.

### 2.2. Field sampling

Two edge sites at each location were sampled; we sampled at SDNWA from June 25–28, 2017 and sampling at CLC took place June 29 –July 6, 2017. At each edge site (n = 2 per location), three transects were laid perpendicular to the grassland-cropland edge and spaced 3 meters apart. Along each transect, samples were taken at the edge (0 m), 25 cm, 50 cm, 1 m, 2 m, 6 m, 8 m, 16 m, and 33 m into each of the two land use types (n = 15 per transect, 90 per location). Each sampling point was randomly assigned a position directly on the transect, or 1 m to either side of the transect (Fig 1). The edge point was visually determined, aided by inspecting the seeding row extent.

At each sampling point, percent cover was assessed for all plant species within a 1 m$^2$ quadrat and 1 m$^2$ quadrats were not allowed to overlap between sampling points. We also recorded plant species present within a 1 m radius of the center point; a 1 m radius was chosen to capture plants whose roots may be in the locale of the soil sample. Aboveground biomass was collected in a 20 cm x 50 cm quadrat and separated into three categories: grass, forbs, and plant litter. Biomass samples were dried at 40°C for four days and weighed to determine dry biomass. During analyses, we combined forbs and grass to encompass all living biomass. To characterize soil properties, we collected a soil core (5 cm diameter x 10 cm depth) from the A horizon at the center of the cover quadrat using a sledge core (AMS Soil Core Sampler, American Falls, ID). A composite sample of three smaller cores (2 cm diameter x 15 cm depth each) was collected for molecular analysis of the soil microbial community. All soil samples were stored at -20°C and were freshly thawed prior to analysis.

### 2.3. Soil property analyses

Soil was air-dried and passed through a 5 mm sieve to remove large debris and rocks. Soil nitrate ($NO_3$) and ammonium ($NH_4$) extractions were performed using 2.0 M KCl [48] and

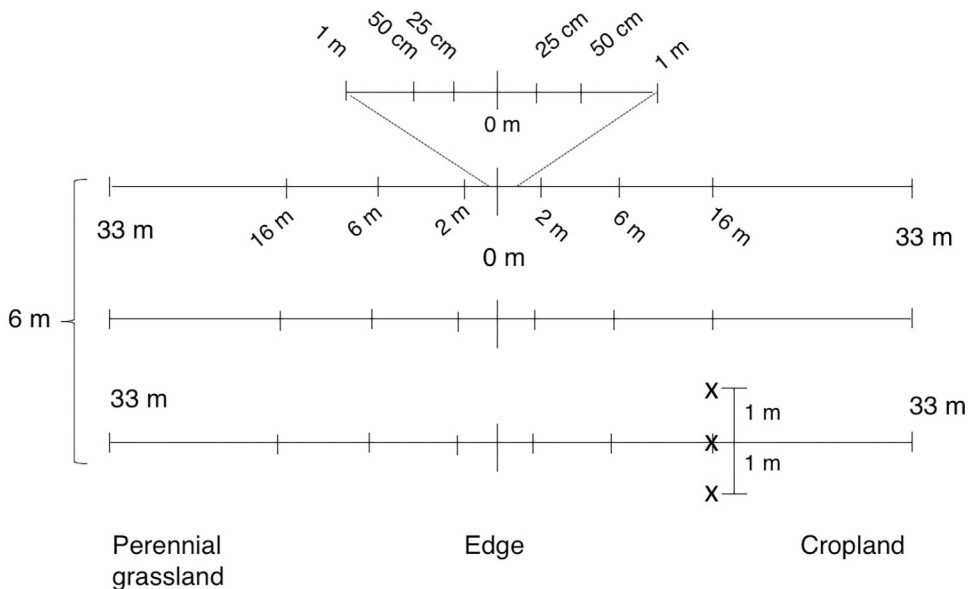

**Fig 1. Transect sampling design.** Transect sampling design at perennial grassland-annual cropland edges. At each edge site, two per location, three transects (33m from edge into each land use) were spaced 3m apart. Each transect had 15 sampling point locations relative to the edge (25cm-33m). Each sampling point along the transect was randomly assigned to one of three positions (x): 1m left, 1m right or on the transect. At each site, there was a total of 90 sample points (2 sample locations * 3 nested transects * 15 sampling points per transect).

analyzed on an AutoAnalyzer 3 (SEAL, UK). Soil pH was measured with a pH probe (Mettler Toledo, USA) using a 1:2 soil to 0.1 M $CaCl_2$ solution [49]. Air-dried, sieved soil was ball-ground (Retsch MM-400, Germany) and 0.25 g of soil was used to determine total N and C. Total C was combusted at 1100°C with a LECO C632 analyzer (LECO, USA) and total N was combusted at 1250°C with the TruMac CNS analyzer (LECO, USA).

## 2.4. Soil microbial sequencing and bioinformatics

Composite samples were sub-sampled (5 g) and ball-ground (Retsch MM-400, Germany). DNA was extracted from 1 g of soil using the PowerPlant Pro Kit (Qiagen, Germany) and eluted in 100 μL of EB solution. DNA was quantified using the Qubit 2.0 Fluorometer (Invitrogen, Massachusetts, USA) and all samples were standardized to 1 ng/μL of DNA for downstream amplification.

To target the bacterial community, the 16S rRNA V4 region was amplified using the primers 515F/806R [50]. Reactions were performed at a final volume of 25 μL; 2 μL of template DNA, 12.5 μL of Platinum Green (2X) Master Mix (Thermo Fisher, Massachusetts, USA), and 1.5 μL of each primer (10 μM). PCR conditions followed Caporaso et al., (2011) [50] using 30 cycles. To target the fungal community, the Internal Transcribed Spacer (ITS) region was amplified using the primer pair ITS1-F [51] and ITS2-R [52]. Reactions were performed at a final volume of 25 μL; 2 μL of template DNA, 12.5 μL of Platinum Green (2X) Master Mix (Thermo Fisher, Massachusetts, USA), 1 μL of each primer (10 μM). PCR conditions were 3 minutes 94°C, 35 cycles: 94°C 30 s, 52°C 30 s, 72°C 45 s, and 72°C for 7 minutes.

All PCR products were purified using the NucleoMag NGS Clean-up and Size Select magnetic beads (Macherey-Nagel, Germany) following the protocol for single size selection with the exception of reduced drying time after the second ethanol wash (2 minutes). Double size selection purification was performed for the ITS amplicon to ensure that fragments larger than

the target region were removed. Library preparation for Illumina MiSeq followed the Illumina Library Preparation Guide (#15044223 Rev. A) and sequencing was performed at the Toxicology Centre at the University of Saskatchewan (300 cycle v2 kit for 16S, 500 cycle v2 kit for ITS).

Soil microbial sequences were processed through QIIME2 2018.11 [53] using the DADA2 pipeline [54]. DADA2 was used for quality filtering, removal of chimeric variants, and merging forward and reverse ITS reads (only forward reads were used for 16S sequences due to poor overlap). Taxonomy was assigned to Amplicon Sequence Variants (ASVs) using the Greengenes [55] and UNITE [56] databases for 16S and ITS, respectively.

## 2.5. Statistical analyses

All statistical analyses were conducted in R 3.5.2 [57]. We performed a non-metric multidimensional scaling (NMDS) analysis on plant species cover at each site using the *vegan* package v 2.5–2 [58]. Plant cover data were Hellinger transformed prior to the NMDS [59]. Soil property vectors overlaid on the NMDS were created using the 'envfit' function in *vegan*. From the NMDS, three groups based on sampling point location were apparent. Thus, we split sampling points into three edge locations: perennial grassland, edge, and cropland (n = 5 per transect, n = 30 for each edge location per site). Perennial grassland and cropland included sampling points from 1 m– 33 m on either side of the edge. Edge included samplings points at 0 m, 0.25 m, and 0.5 m into both perennial grassland and cropland. The groupings were examined by permutational multivariate analysis of variance (PERMANOVA) using the adonis function in the *vegan* package. Indicator plant species for each edge location were determined with the *indicspecies* package v 1.7.6 [60].

To examine vegetation biomass and soil properties across the edge, we used linear mixed models (LMM). Fixed effects for all models included edge location, site, and their interaction. Random effects included transect (n = 3) nested within site (n = 2). Total living, grass, forb, and litter biomass, as well as $NO_3$ and $NH_4$ were log transformed to meet assumptions of normality. Models were fit with the *lme4* package v 1.1–19 [61] using restricted maximum likelihood (REML) estimation. Model fit was assessed by inspecting residuals to ensure homoscedasticity. We used the *lmerTest* package v 3.0–1 [62] to obtain degrees of freedom and *p*-values. Tukey's HSD post-hoc testing was used to determine significant differences among edge location using the *emmeans* package v 1.3.1 [63].

We conducted an NMDS to examine the bacterial and fungal community of each site. Again, we used a Hellinger transformation on the ASVs, as it places less weight on rare species [64]. The previously established three groups were also examined for the bacterial and fungal communities by PERMANOVA using the adonis function in the *vegan* package.

We used Structural Equation Models (SEMs) to investigate relationships between land management, plants, soil properties, and the soil microbial communities. An advantage of using SEMs is the ability to include multiple complex relationships in an *a priori* theoretical model [65]. Our *a priori* SEM hypothesized that land management had a direct relationship with plants (live plant biomass) (Fig 2). Land management affects plant biomass through direct manipulation of plant community via seeding, harvesting, and mowing. Plant biomass was log-transformed to improve linearity. As land management was included as a categorical variable with three factors (cropland, edge, and perennial grassland), we ran the SEM twice, changing the reference land management category to display all possible comparisons (cropland vs edge, perennial vs edge, cropland vs perennial). We also hypothesized a direct relationship from plant biomass to total C and total N as studies show biomass is an important factor [66, 67]. Lastly, we included an effect of soil properties on the fungal and bacterial communities as soil nutrients may influence soil microbial communities [68, 69].

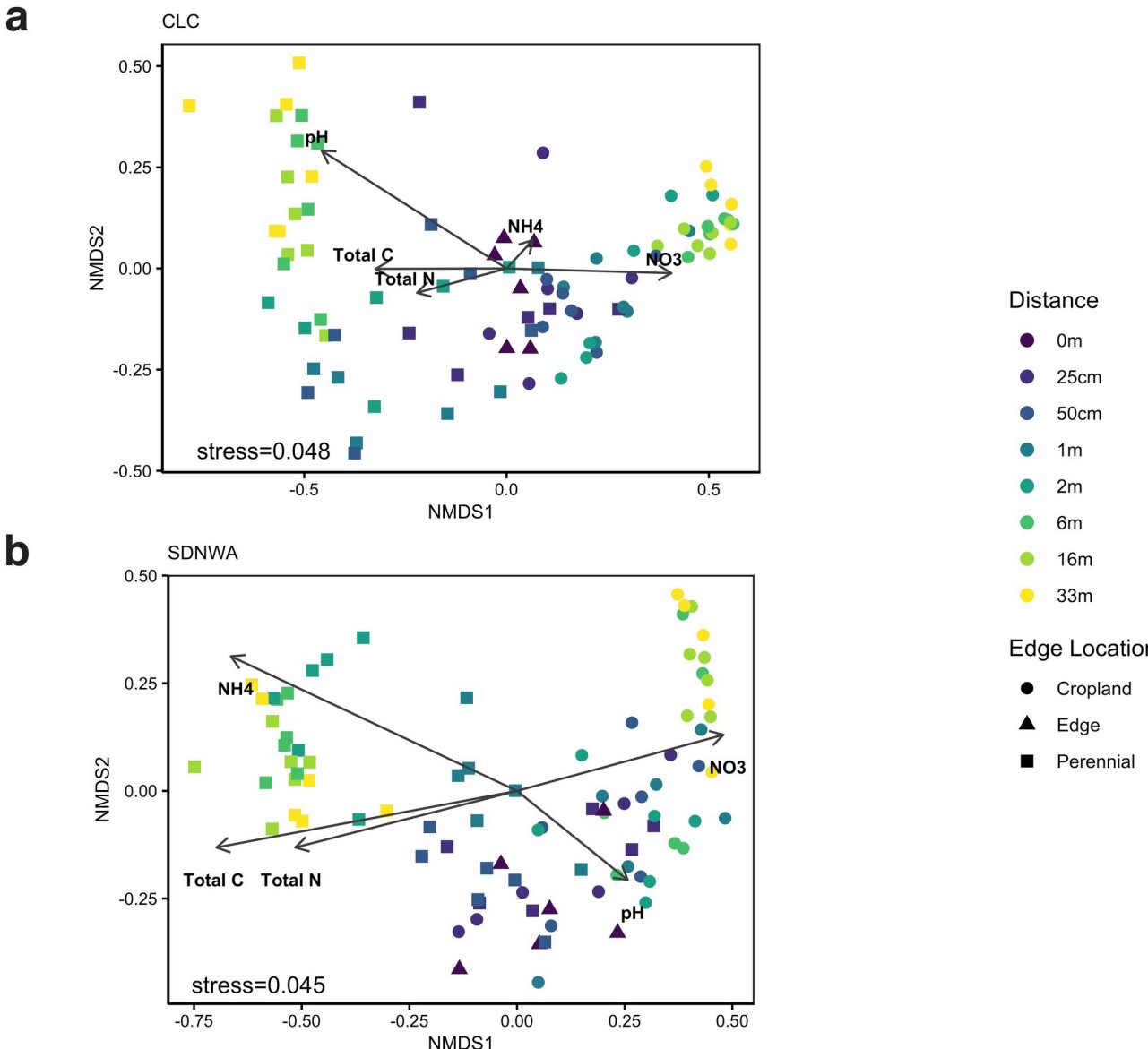

**Fig 2. Non-metric multidimensional scaling analysis of vegetation cover.** A non-metric multidimensional scaling analysis of vegetation cover at (a) Conservation Learning Centre and (b) St. Denis National Wildlife Area. The colour gradient represents sampling points from 0 m to 33m (into either cropland or perennial grassland), with 0 m being the edge. Shapes represent edge location, triangles are edge points (0m), squares represent points in the perennial grassland and circles represent points in the cropland. Soil property vectors are overlaid on each plot.

Goodness of fit for SEMs was assessed by the chi-square ($p$-value > 0.05), Root Mean Square Error of Approximation (RMSEA < 0.08), and Comparative Fit Index (CFI > 0.90) [70, 71]. As our initial *a priori* model was not a good fit ($\chi^2$ $p$-value < 0.001, RMSEA = 0.299, CFI = 0.692), we evaluated alternative models [72]. As such, our modelling approach was now exploratory and based on modification indices we added pathways with ecological relevance [73]. All models were fit and calculated using the *lavaan* package v 0.6–3 with the maximum likelihood estimation [74].

To further investigate the fungal community, we identified significant fungal genera across the edge at both sites. First, the ASV table was filtered at 20% prevalence across samples to

remove rare species and to prepare data for transformation, zero and NA values in the ASV tables were replaced with an estimate (Count Zero Multiplicative) using the zCompositions package [75]. The centered log-ratio transformation was calculated with the CoDaSeq package v 0.99.4 [76] and these ratios were used for abundance. Genera were aggregated using the *phyloseq* package v 1.24.1 [77] and Welch's *t*-tests used to determine significant differences in genus abundance between each pair of edge location (cropland vs edge, edge vs perennial, perennial vs cropland). *P*-values were adjusted using the p.adjust function in R selecting Bonferroni correction method.

## 3. Results

### 3.1. Vegetation community and biomass

Differences in plant community composition were strongly related to edge location (Fig 3). Three distinct clusters were identified: the edge (0.5 m-0.5 m), the cropland (33 m-1 m), and the grassland (1 m-33 m) at both CLC and SDNWA. These plant communities across the edge appear to correlate with soil properties (Fig 2).

The distinct vegetation groupings for edge, perennial grassland, and cropland were driven by abundant non-native annual plant species at the edge, seeded species in the perennial grassland, and the crop in the croplands. Indicator species at the edge included hemp nettle (*Galeopsis tetrahit* L.) and cleaver's (*Galium aparine* L.) at both sites (S1 Table). Non-native annual and some perennial plant species, here called weedy species, were dominant at the edge and comprised 77% ± 8.9% (mean ± SD) of edge plants recorded at CLC and 85% ± 7.4% at SDNWA. In perennial grasslands, *B. inermis* had the highest indicator value of any species at SDNWA, while at CLC, both *B. inermis* and *B. bieberstenii* were strong indicator species. Other indicator species for perennial grassland common to both sites included *M. satvia* and dandelion (*Taraxacum officinale* L.). Indicator species for cropland were the crops planted in 2017, *B. napus* and *L. usitatissimum* for CLC and SDNWA, respectively.

Patterns of aboveground vegetation biomass across the edge differed at each site; as determined by linear mixed modelling, the interaction was significant between site and edge location for each biomass category. At SDNWA, living biomass was greatest in the grassland and significantly decreased across the edge and cropland, however at CLC living biomass was only significantly higher in the perennial grassland compared with the edge (S2A Fig). The greatest forb biomass at CLC was in cropland, due to planted canola, while at SDNWA the greatest forb biomass was at the edge and cropland (S2B Fig). At the edge, forbs consisted of 74% ± 31% and 88% ± 23% (mean ± standard deviation) of living biomass at CLC and SDNWA, respectively. Not surprisingly, the majority of grass biomass was in the perennial grasslands (S2C Fig). Litter biomass was not significantly different across the edge at either site (S2D Fig).

### 3.2. Soil properties

Overall, soil properties changed across the edge; however, the pattern for total C, $NH_4$, and pH significantly differed between sites (S2 Table). Total C and N were significantly higher in the perennial grasslands than croplands at both sites. At SDNWA, the edge had intermediate levels of total C and N when compared to grassland and cropland; at CLC, total C and N at the edge were more similar to croplands (S3A and S3B Fig). $NO_3$ had the opposite trend as total C and N, with significantly higher values in the cropland and edge than the perennial grassland at both sites (S3C Fig). SDNWA had significantly higher $NH_4$ in perennial grassland compared to edge and cropland, while at CLC, $NH_4$ was similar across all locations (S2 Table). Soil pH was significantly higher in the perennial grassland at CLC compared to edge and cropland, with pH values ranging across the edge from 4.8–6.9 (S3E Fig). At SDNWA, pH was not

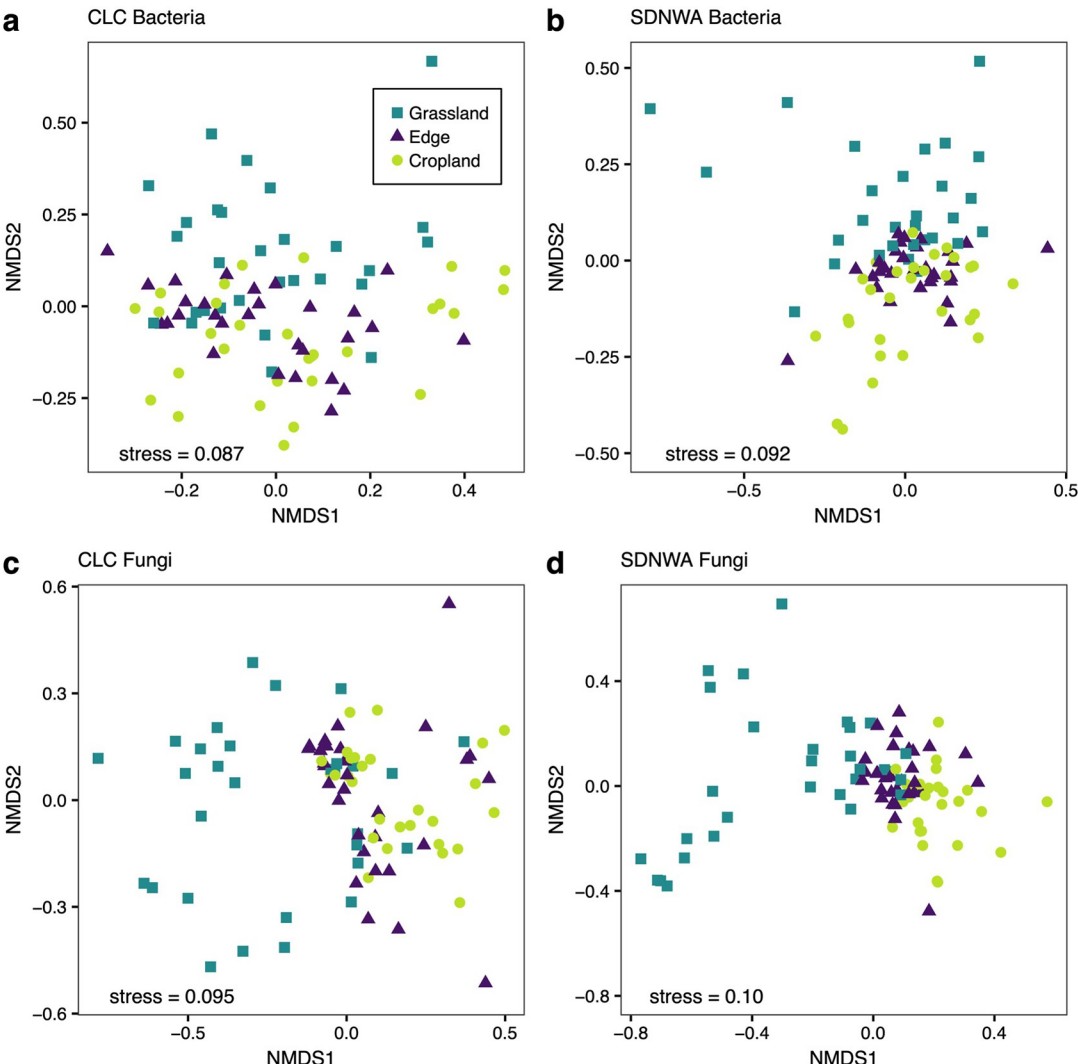

**Fig 3. Non-metric multidimensional scaling analysis of microbial community across croplands and perennial grasslands.**
Non-metric multidimensional scaling analysis for the (a, b) bacterial community and the (c, d) fungal community at the
Conservation Learning Centre (CLC) and St. Denis National Wildlife Area (SDNWA). Shape and colour of the points represent
location across the edge; perennial grassland (teal squares), edge (purple triangles), and cropland (yellow circles).

significantly different across the edge, with values that ranged from 6.5–7.5. Overall, soil prop-
erties at edge locations were more variable at CLC than at SDNWA.

### 3.3. Soil microbial community

Both bacterial and fungal communities were different across the edge at CLC and SDNWA
(PERMANOVA, S3 Table) (Fig 3). Changes in the bacterial community were less clear, how-
ever; at SDNWA, bacterial community composition appears to diverge more with respect to
edge location than at CLC (Fig 3A and 3B). Fungal communities at both sites appeared to have
a distinct perennial grassland community compared with the edge and cropland (Fig 3C and
3D).

### 3.4. Structural equation modelling

Our final SEMs, after including direct pathways from land management to both soil properties and microbial communities, were a good fit, with edge as reference ($\chi^2$ p-value = 0.144, RMSEA = 0.074, CFI = 0.996) and perennial grassland as reference ($\chi^2$ p-value = 0.144, RMSEA = 0.074, CFI = 0.996) (S4 Table). We were able to explain 36% of the variation in the fungal community, which was driven primarily by land management (Fig 4). Cropland had a 'positive' relationship and perennial grasslands a 'negative' relationship with the fungal community when compared to the edge, indicating community composition differences (Fig 4A); both cropland and edge had 'positive' relationships with the fungal community, when compared to perennial grasslands (Fig 4B). Therefore, the fungal community was most strongly positively influenced by the cropland, followed by edge, and negatively influenced by perennial grasslands. Bacteria was similarly affected by the cropland and edge (Fig 4A) and had a 'negative' relationship with cropland and edge, compared with perennial grasslands (Fig 4B). Plant biomass had no significant relationships with soil properties but soil properties were significantly influenced by land management (Fig 4A and 4B). Perennial grasslands had 'positive' relationships with both total C and N compared to the edge, while cropland had a 'negative' relationship with total C (Fig 4B). These finding are supported by the significantly higher TC and TN detected in the perennial grassland and the edge having intermediate TC at SDWNA. Similarly, land management relationships with plant biomass follow the same pattern we observed from the linear mixed effect model; the greatest plant biomass was in perennial grasslands (Fig 4A), followed by edge, and then cropland (Fig 4B). While land management had direct impacts on the soil microbial community, soil properties and plant biomass, we did not find any significant pathways from soil properties to the microbial communities. In addition, the interaction between the fungal and bacterial communities was not significant in either model (Fig 4A and 4B).

### 3.5. Fungal abundance across the edge

Since changes in the fungal community were clearly related to land use (Fig 4) and these differences were more distinct at both of our sites (Fig 3), we further examined shifts in fungal community composition across land uses. After filtering the data set to obtain the most abundant genera (see methods), 50 genera remained (from 392) and six genera were found to be significantly different between at least one location comparison (i.e. cropland vs perennial, edge vs grassland, edge vs cropland). The abundances of five out of the six genera were significantly greater in the cropland than the grassland (Table 1). Two of these genera, *Clonostachys* and *Gibberella*, were also found in greater abundances at the edge compared to the grassland. *Paraphoma* was the only genera that was significantly more abundant at the edge than in the cropland (Table 1).

## 4. Discussion

We investigated soil properties, vegetation community, and the soil microbial community across edges of perennial grasslands and annual croplands. Land management had direct and indirect influences on the soil microbial community through changes in vegetation and soil properties. Edges acted as an intermediate and unique environment between the two land uses, composed of predominately non-native weedy plants and the edge was more similar to cropland than grassland in both plant and soil.

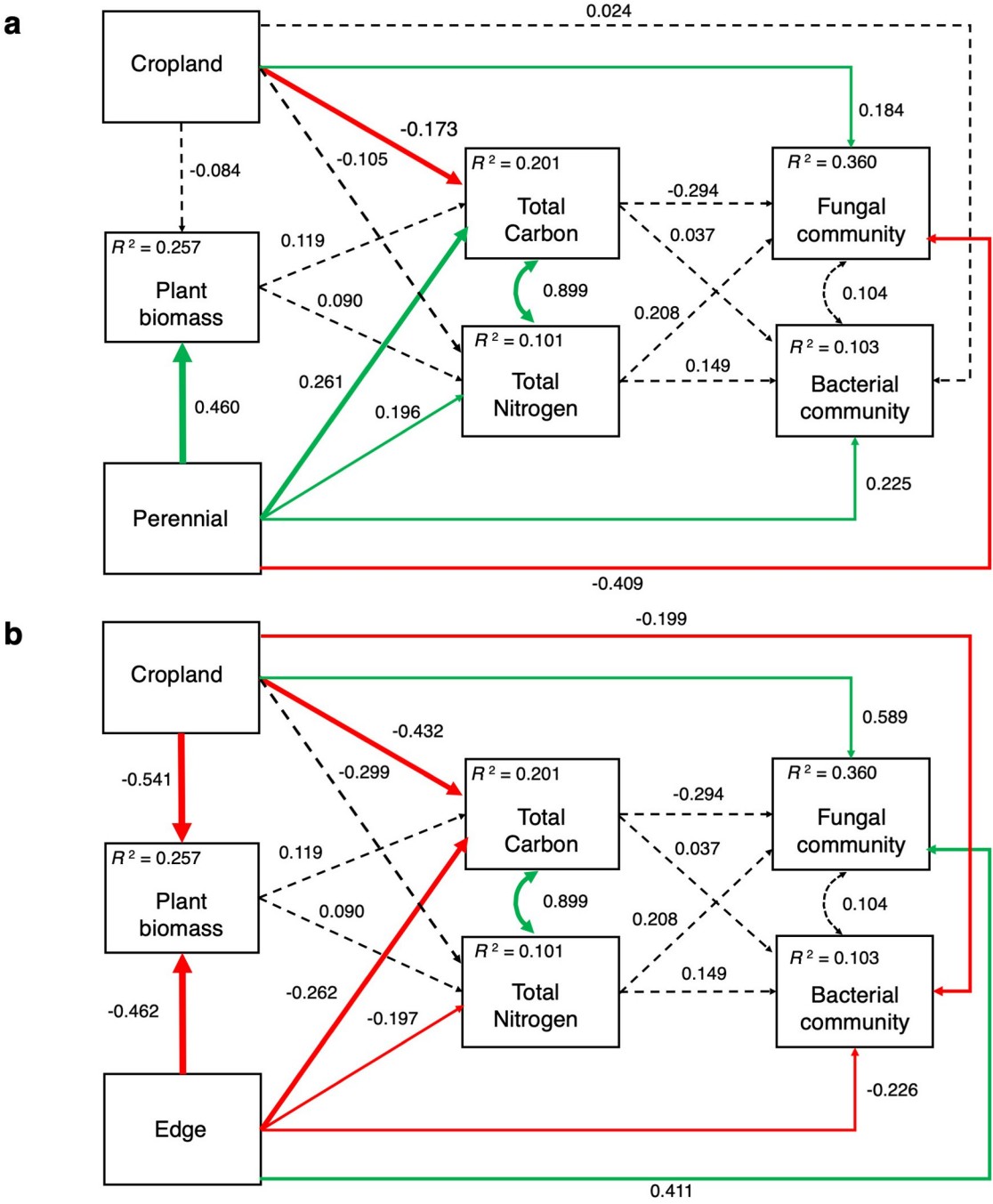

**Fig 4. Structural equation model for relationships between land management, microbial communities, soil properties, and vegetation.** Final structural equation model ($\chi^2$ p-value = 0.144, RMSEA = 0.074, CFI = 0.996) representing the causal relationships between land management, aboveground plants, soil properties and microbial communities with edge (a) and perennial (b) as the reference category. Data from both SDNWA and CLC are included. Solid arrows are significant (p < 0.05) and pathways with dashed arrows are non-significant. Green arrows represent significant positive pathways and red arrows represent significant negative pathways. Standardized partial path coefficients are beside each pathway arrow and R$^2$ values provided for each dependent variable.

**Table 1. Fungal genera abundance across croplands and perennial grasslands.**

| Class | Order | Family | Genus (*p*-value) | Abundance |
|---|---|---|---|---|
| Dothideomycetes | Pleosporales | Phaeosphaeriaceae | *Chalastospora* (0.001[a]) | C > E > G |
| Dothideomycetes | Pleosporales | Phaeosphaeriaceae | *Clonostachys* (0.002[a], <0.001[b]) | E > C > G |
| Sordariomycetes | Hypocreales | Nectriaceae | *Gibberella* (<0.001[a], <0.001[b]) | C > E > G |
| Dothideomycetes | Pleosporales | Phaeosphaeriaceae | *Paraphoma* (0.002[c]) | E > G > C |
| Dothideomycetes | Pleosporales | Phaeosphaeriaceae | *Parastagonospora* (0.001[a]) | C > E > G |
| Sordariomycetes | Hypocreales | Nectriaceae | *Sarocladium* (0.005[a], 0.002[c]) | C > E > G |

Fungal genera with significantly different (p < 0.05) abundances between edge location (at least one significant difference between [a] cropland-grassland, [b] edge-grassland, [c] edge-cropland). Significance was determined by Welch's t-test using abundance values from both sites after centered log-ratio transformation to obtain compositional abundance. All genera were in the Ascomycota phyla and Pezizomycotina subphyla. The order of abundance (greatest to least) is indicated in the abundance column (C = Cropland, E = Edge, G = Grassland).

## 4.1. Aboveground changes across the edge

Differences in plant community composition and biomass across the edge was largely determined by land use type. Three different vegetation communities were observed: the perennial grassland, the edge (~1 m in width), and the cropland. Unsurprisingly, cropland vegetation was strongly influenced by the crop seeded; *B. napus* at CLC and *L. usitatissimum* at SDNWA. Living biomass was greatest in grasslands, which were dominated by brome species (*B. inermis* and *B. biebersteinii*) that were seeded in previous years. Both brome species were primary contributors to biomass, as grass constituted 88% of total living biomass.

Plant community composition at the edge was a mixture of grassland plants, crops, and weedy species. Weed population densities are highest near, or at, an edge [78] because these plants are disturbance tolerant [79]. Non-native plant presence in agriculture frequently increases plant species richness in these settings and is driven by agronomic activities [80, 81]. Agronomic activities including general mechanical disturbance such as mowing, crop sowing, and harvesting disturb the edge [82]. While our study sites were no-till systems, croplands still experienced a higher level of disturbance than grasslands throughout the growing season. In-field herbicide and fertilizer application can have unintended effects on adjacent areas [83]. Herbicide and fertilizer drift can reach beyond cropland edges and affect the plant community [84, 85]; for example, fertilizer drift can promote faster growing competitive plant species that outcompete others [84, 86, 87]. In addition to higher nutrient availability, cropland edges have open space allowing undesirable weedy species to establish [82, 88]. These edge effects lend advantages to these plant species that may compete with crops, reducing yields [89] and facilitate invasion of undesirable plants into adjacent, more natural, land use types [90].

Management practices, such as using herbicides or doubling sown crop density are effective in reducing weed populations at edges [91]. However, conventional eradication attempts may bring more detriments to larger agroecosystem, herbicide can drift into non-target areas and weedy species can become herbicide resistant [92]. Field edges can act reservoir for invasive weeds and other undesirable microbial pathogens [93]. However, the reverse is also true, a diverse weed community can provide ecosystem services and habitat to beneficial species [82, 94, 95]. Multiple management strategies are needed to successfully manage edge habitats valuable to many aspects of the agroecosystem.

## 4.2. Belowground changes across the edge

Land management practices indirectly influenced soil physiochemical properties across perennial grassland-cropland edges through modification of aboveground plant community, and

directly through fertilizer application. We found total C and N were highest in the perennial grasslands and lowest in the cropland; this is common in agroecosystems as soil quality is often poorer in cultivated land compared to non-cultivated land [96–98]. At our sites, perennial grasslands had plant species with relatively high-quality litter that likely influenced soil properties through the deposition of rich C sources. For example, at our sites in the perennial grasslands, *B. inermis* and *M. stavia* produce large amounts of litter that quickly degrades and is high in N content with a low C:N, which can increase soil organic C and rates of soil N cycling [99–101]. In addition, while the cropland is relatively productive, the majority of aboveground biomass is removed, not allowing the plant based C to return to the soil, which is a major source of soil C [102].

Edges are subjected to fertilizer applied to the cropland, evidenced by high spikes of $NO_3$ in both in cropland and edges. Inorganic N amendments, applied over both long and short time periods, can increase soil total N and $NO_3$ [103–105]. Nitrate concentrations in edge soils were more similar to croplands, likely due to the close proximity of the edge to the cropland and inputs from surface runoff [106]. However, our observation was only at one time point and may not provide a complete picture of N dynamics and seasonal fluctuations of $NO_3$ in this system. Regardless, edges in agroecosystems appear to act as a buffer for nutrient movement from managed croplands into adjacent land use types.

## 4.3. Soil microbial community across the edge

In our study, land management appeared to have a strong influence on soil microbial community composition, as the direct pathways from land management to microbial communities were mostly significant in the SEMs. We chose to focus on community composition rather than a metric like richness, because in cases where richness is not affected, composition can detect more discreet changes [2, 107]. Management practices can directly and indirectly affect soil microbial communities [108–110] and long term practices have selective forces on the soil microbial community, thus changing the microbial community composition as it adapts to these disturbances [111]. Fungal community composition was different in the grassland than cropland, as denoted by a 'negative' impact by the perennial grassland and a 'positive' by the cropland and edge. Fungal community composition was also different between the edge and cropland, though not as pronounced. Bacterial community composition was also different in the perennial grasslands compared to edge or the cropland, however patterns of response across the land uses were not as clear for bacteria as fungi (Fig 3). Bacterial communities may respond less than fungal communities to changes in land use and vegetation, similar patterns were found in no-till cropland and native prairie in Kansas [26] and in comparing native and exotic grasslands [112]. Direct relationships between land management and the microbial community is likely driven by underlying changes of soil and plants associated with land use types.

Plants are an important factor affecting microbial communities, especially at our study sites, land management created three distinct plant communities across the edge. Plant species can influence soil microbes through symbiotic relationships, root exudates, and plant litter inputs [112]. A key difference in plant community across the edge was the dominance of annual plants in the cropland and edge, while the grassland was composed of nearly all perennial plants. *Brassica* species, like the *B. napus* planted at CLC are non-mycorrhizal plants, which would greatly affect both the quantity and quality of AMF hyphae and spores observed [113, 114], thus could be an aspect shaping fungal community composition. The distinction between annual and perennial plants is important as McKenna et al., (2020) [113] found that soil fungal community composition was similar under two different perennial vegetation types

a seeded monoculture of intermediate wheatgrass (*Thinopyrum intermedium* (Host) Barkworth & D.R. Dewey) grassland and a native prairie. However, both perennial fungal communities were different than the fungal community under annual crop rotation. Root architecture and activity may be largely responsible for differences between annual and perennial plants, as perennial grasslands have greater root biomass and more evenly distributed and deeper roots than annual croplands [26]. Annual plants dominated the cropland and edges, which had similar direct effects on fungal community (Fig 4A), suggesting that the life history strategies of dominant plants influence the fungal community.

Although we did not observe significant pathways from soil nutrients to fungi or bacteria, we did observe a strong influence of land use on soil nutrients. The perennial grassland had more total N likely due to more biomass, but high $NO_3$ and $NH_4$ were observed in the cropland. Fertilizers containing N can reduce fungal diversity and fungal richness, possibly related to $NO_3$ [32]. However, others have found no effect of N fertilizers on fungal diversity or richness [114, 115], but differences in fungi community composition [31]. Increased N availability, specifically $NO_3$, may be disrupting natural plant-soil feedback relationships [31, 116]. By increasing the N available to soil fungi or interrupting available C exudates via N available to plants, $NO_3$ can alter community composition by promoting or suppressing fungi with different life history strategies based on altered soil conditions [104, 117]. Higher $NO_3$ levels in the cropland and edge may have been an important driver of microbial community composition, specifically fungi at our study sites. One aspect not considered directly in this analysis, was the soil C to N ratio. The C:N is crucial for microbial functioning [118, 119] and linked to soil microbial community composition [120–122]. Considering the soil C:N explicitly in the future would aid in understanding soil microbial community composition across the edge.

Examining abundant fungal genera revealed further insight into the effect of land management on the fungal community. Plants and soil fungi often develop a stable environment together as their interactions can provide mutual benefits, such as aid in nutrient acquisition for plants and carbon sources for fungi through plant exudates [118]. Different plant species can affect soil fungi differently, likely due to unique soil microbiomes associated with each plant species [119]. For example, plant species with litter high in C:N can promote Basidomycota fungi to aid in decomposition, changing fungal community composition [120, 121]. Fungal genera *Gibberella* and *Paraphoma* were significantly more abundant at the edge and likely reflect the presence of both crop species and grasses. Many *Gibberella* species are plant pathogens that can cause significant crop diseases, such as head blights in grain crops and ear rot in corn (*Zea mays* L.) [122]. *Paraphoma* are common soil fungi and frequently associate with monocots [123]. Furthermore, at the edge we found *P. chrysanthemicola*, a plant pathogen [124, 125] known to affect plants in the Asteraceae and Rosaceae families [126] which were found at the edge. Significant fungal genera abundant in the cropland were mostly pathogenic, including *Sarocladium* [127] and *Parastagonospora*; *P. nodorum*, a major wheat pathogen, which was identified to the species level [128]. Others have hypothesized that edges can act as a reservoir for undesirable microbial pathogens [93]. In our study the difference between fungal communities in cropland and edges, compared to perennial grasslands, was driven by the abundance of pathogens in these more heavily managed land uses supporting this hypothesis.

## 5. Conclusions

In our study, we saw differences across the edge aboveground and belowground; changes included plant community composition, soil total N and C, and soil microbial community composition. Aboveground, weedy species were most abundant at the edge and appeared to have a positive response to the edge, where conditions from the cropland and grassland made

it ideal for those species [107]. Belowground, soil C and N were lowest in the cropland, but $NO_3$ was highest in the cropland and edges. Soil microbial community composition across the edge was different, and fungi had more apparent differences in community composition than bacteria. A more in-depth analysis on fungi, showed some genera were more abundant in the cropland, edge, or grassland. For a holistic understanding of agroecosystem impacts, future studies need to consider the interrelated effects of management on soil properties and plant communities as these factors are often driving changes in soil microbial communities [110, 129]. Further knowledge of the interactions between the soil microbial community, soil properties, plants, and edges in the agroecosystem will help to develop more sustainable agricultural practices and build healthier more resilient agroecosystem.

*Raw sequence fasta files and the associated metadata can be found at the National Center for Biotechnology Information (NCBI) under Bioproject PRJNA588061

## Supporting information

**S1 Fig. *A priori* model used for structural equation models.** A priori model used for structural equation modelling. Direct relationships are represented by straight arrows and curved arrows represent unexplained covariate relationships. The first and second axes from nonmetric multidimensional scaling analyses was used to represent the fungal and bacterial communities.
(TIF)

**S2 Fig. Biomass across croplands and perennial grasslands.** Aboveground vegetation biomass (dry weight g/m2) across edge locations (perennial grassland (dark grey), edge (light grey), and cropland (white) at the Conservation Learning Centre (CLC) and St. Denis National Wildlife Area (SDNWA). Boxes encompass 25–75% quantiles of the data, while whiskers encompass 5–95%. The median is indicated by the black horizontal line, and outliers are shown as dots. Different letters indicate a significant difference (p-value < 0.05) between edge locations determined by Tukey-HSD post-hoc tests on linear mixed models.
(TIF)

**S3 Fig. Soil properties across croplands and perennial grasslands.** Soil properties across edge locations (perennial grassland (dark grey), edge (light grey), and cropland (white) at the Conservation Learning Centre (CLC) and St. Denis National Wildlife Area (SDNWA). Different letters indicate a significant difference (p-value < 0.05) between edge locations determined by Tukey-HSD post-hoc tests on linear mixed models.
(TIF)

**S1 Table. Indicator plant species for each edge location (perennial grassland, edge, and cropland) at the Conservation Learning Centre (CLC) and the St. Denis National Wildlife Area (SDNWA).** Indicator species are also listed with edge + grassland and edge + cropland. Edge + Grassland is the combination of edge and grassland points on the transect, while Edge + Cropland is the combination of edge and cropland points on the transect.
(DOCX)

**S2 Table. F-values (*p*-values) from linear mixed models for biomass (g/m$^2$) and soil properties (total C and total N (%), NH$_4$ and NO$_3$ (µg/g soil), and pH) across edge location (perennial grassland, edge, and cropland), site (Conservation Learning Centre and St. Denis National Wildlife Area), and their interaction.** Significant *p*-values are bolded and log transformed data are denoted by [†].
(DOCX)

**S3 Table. Results from the PERMANOVA for bacteria and fungi at Conservation Learning Centre and St.** Denis National Wildlife Area.
(DOCX)

**S4 Table. Estimate parameters from both final structural equation models.**
(DOCX)

## Acknowledgments

Thank you to Alix Schebel for assistance in the laboratory; and Megan Horachek, Raidin Brailsford, Lauren Reynolds, Lukas Smith, and Trang Nguyen for their help with field sampling. Lastly, thank you to Steve Mamet for providing a review of the manuscript.

## Author Contributions

**Conceptualization:** Katherine J. Stewart.

**Formal analysis:** Mariah Aguiar, Alexandra J. Conway.

**Funding acquisition:** Katherine J. Stewart.

**Investigation:** Mariah Aguiar.

**Methodology:** Jennifer K. Bell.

**Supervision:** Katherine J. Stewart.

**Validation:** Jennifer K. Bell.

**Writing – original draft:** Mariah Aguiar.

**Writing – review & editing:** Alexandra J. Conway, Jennifer K. Bell.

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
