## [Decision Letter · Decision Letter 0]

9 Nov 2022

PONE-D-22-22877Living on the edge: fungal community strongly influenced by land management at perennial grassland-cropland edgesPLOS ONE

Dear Dr. Stewart,

Thank you for submitting your manuscript to PLOS ONE. After careful consideration, we feel that it has merit but does not fully meet PLOS ONE’s publication criteria as it currently stands. Therefore, we invite you to submit a revised version of the manuscript that addresses the points raised during the review process.

We look forward to receiving your revised manuscript.

Kind regards,

RunGuo Zang

Academic Editor

PLOS ONE

Journal Requirements:

“This research was supported by a Natural Sciences and Engineering Research Council of Canada Strategic Partnership Grant entitled “Understanding resilience in agroecosystems: landscapes in transition" with Agriculture and Agri-Food Canada. Thank you to Alix Schebel for assistance in the laboratory; and Megan Horachek, Raidin Brailsford, Lauren Reynolds, Lukas Smith, and Trang Nguyen for their help with field sampling. Lastly, thank you to Steve Mamet for providing a review of the manuscript.”

“This research was supported by a Natural Sciences and Engineering Research Council of Canada Strategic Partnership Grant entitled “Understanding resilience in agroecosystems: landscapes in transition" with Agriculture and Agri-Food Canada. The funders had no role in study design, data collection and analysis, decision to publish, or preparation of the manuscript.”

Additional Editor Comments:

Although the two referees have some concerns on the manuscript,they are generally positive to it.You can refer to the comments of the refees and revise the manuscript accordingly.

Reviewers' comments:

Reviewer's Responses to Questions

**Comments to the Author**

1. Is the manuscript technically sound, and do the data support the conclusions?

Reviewer #1: Partly

Reviewer #2: Partly

2. Has the statistical analysis been performed appropriately and rigorously? 

Reviewer #1: Yes

Reviewer #2: N/A

3. Have the authors made all data underlying the findings in their manuscript fully available?

Reviewer #1: Yes

Reviewer #2: Yes

4. Is the manuscript presented in an intelligible fashion and written in standard English?

Reviewer #1: Yes

Reviewer #2: Yes

5. Review Comments to the Author

Reviewer #1: Edge effects in agroecosystems is crucial for maintaining ecosystem function. This study examine impacts of land management on aboveground and belowground edge effects, which would help our better understanding of Edge effects .

Title: Living on the edge might be unnecessary. What kinds of fungal community? Unclear. What and how about the land management? Could you be specific? I think perennial grassland-cropland also too large, which place? global of national?

Abstract: It reports "soil fungal and bacterial community composition", so I think the title using fungal community might not cover all subjects the authors conducted.

"Soil total nitrogen and carbon significantly decreased across the edge, " it might need P value and R2 for it, how to prove "significantly"?

Line 39: Can you be specific what "properties" here?

Line 60: Can you list the references for each of the four "essential mechanisms"? All the essential mechanisms from the same reference [6]?

Figures，better put the figure into the text where the legends are so that reviewers can read the text and figures at the same time. No need to go and forth from the start to the end. Each figure should stand independently. Better with different color for all figures.

Figure 1, lines in the sampling are not clear, why three 33 m lines existed? for control? Each 33 m line are conducted the same sampling as the upper one?

Figure 2, why there are no values in the structural equation models? All are positive?

I suggest combine Figure 3 and 6 together? Figure 4 and 5 into one plate, if not, could one of them putting in the appendix? Seven figures are too many I think. Figure 7 is fine, might put it as appendix for Figure 2?

Reviewer #2: This manuscript investigated the plant and soil communities along the grassland-cropland edge, generally the experiment was well designed, however, I am cofused about the analysis about different sites, for the organizing of the results and figures. As you selected two sites to represent the edges, shall they be considered as replications? why you analyze them individually? The three transect in each site was not real replicate. So I think the authors should considered carefully about their experiment design and analyze the data properly.

The figures should be improved, especially for Figure 1 and 2, could be combined as one composite figure.

Another concerns is about the soil parameters, why you only select soil C and N to link with the microbial communities? have you think about the C/N ratios?

6. PLOS authors have the option to publish the peer review history of their article (what does this mean?). If published, this will include your full peer review and any attached files.

Reviewer #1: **Yes: **Hua-Feng Wang

Reviewer #2: No

---

## [Author Response · Author response to Decision Letter 0]

26 Dec 2022

Manuscript PONE-D-22-22877

Dear Editor, 

Thank you for the opportunity to revise our manuscript formerly entitled Living on the edge: fungal community strongly influenced by land management at perennial grassland-cropland edges (revised title: Agroecosystem edge effects on vegetation, soil properties, and the soil microbial community in the Canadian prairie). The reviewer’s comments and feedback were valuable and helpful in improving the manuscript. We have carefully read all comments and have revised the manuscript based upon the reviewer’s suggestions. Below we provide our responses to each comment in italics, and we have uploaded a revised manuscript with tracked changes as requested. Please also note that we have now uploaded our data to Federated Research Data Repository with the following DOI: https://doi.org/10.20383/103.0623. We hope you will now find our manuscript suitable for publication in PLOS ONE. 

Regards, 

Dr. Katherine Stewart

Editor Comments

Comment 1:

The PLOS ONE style requirements were reviewed, and we have ensured that the manuscript meets them, including the file naming requirements. 

Comment 2:

In your Methods section, please provide additional information regarding the permits you obtained for the work. Please ensure you have included the full name of the authority that approved the field site access and, if no permits were required, a brief statement explaining why.

Information regarding the authorization for the field sampling was added in the Methods section. The field sampling was authorized by the St. Denis National Wildlife Area and the Conservation Learning Centre. 

Comment 3:

“This research was supported by a Natural Sciences and Engineering Research Council of Canada Strategic Partnership Grant entitled “Understanding resilience in agroecosystems: landscapes in transition" with Agriculture and Agri-Food Canada. The funders had no role in study design, data collection and analysis, decision to publish, or preparation of the manuscript.”

We have removed funding information from the acknowledgement section. 

Comment 4: We note that you have stated that you will provide repository information for your data at acceptance. Should your manuscript be accepted for publication, we will hold it until you provide the relevant accession numbers or DOIs necessary to access your data. If you wish to make changes to your Data Availability statement, please describe these changes in your cover letter and we will update your Data Availability statement to reflect the information you provide.

The data has been uploaded to a repository, the doi is: https://doi.org/10.20383/103.0623, we have also provided this information in our cover letter.

Comment 5:

References have been reviewed to ensure it is complete and correct. No papers that were cited have been retracted. 

Reviewer 1

Comment 1: 

Title: Living on the edge might be unnecessary. What kinds of fungal community? Unclear. What and how about the land management? Could you be specific? I think perennial grassland-cropland also too large, which place? global of national? 

Abstract: It reports "soil fungal and bacterial community composition", so I think the title using fungal community might not cover all subjects the authors conducted. 

We have reviewed the title and altered it to be more reflective to the study. The title was modified to “Agroecosystem edge effects on vegetation, soil properties, and the soil microbial community in the Canadian prairie”

Comment 2:

"Soil total nitrogen and carbon significantly decreased across the edge, " it might need P value and R2 for it, how to prove "significantly"

The significance was denoted in Supplemental table S2, but we have now also included the P value in the abstract as per your suggestion. 

Comment 3:

Line 39: Can you be specific what "properties" here? 

We have updated the text to specify the soil properties. 

Comment 4:

Line 60: Can you list the references for each of the four "essential mechanisms"? All the essential mechanisms from the same reference [6]? 

The essential mechanisms were discussed at length in the reference [6] as it was the primary paper in which these mechanisms were identified. The authors present a conceptual model of edge dynamics based on previous studies.

Comment 5:

Figures, better put the figure into the text where the legends are so that reviewers can read the text and figures at the same time. No need to go and forth from the start to the end. Each figure should stand independently. Better with different color for all figures.

We agree that figures would be better left in the text for the ease of reading. However, PLOS ONE guidelines state “Do not include figures in the main manuscript file. Each figure must be prepared and submitted as an individual file.”

Figure colours were chosen based on their legibility. All panels within a figure have the same color scheme for coherence as it is the same analyses.

Comment 6:

Figure 1, lines in the sampling are not clear, why three 33 m lines existed? for control? Each 33 m line are conducted the same sampling as the upper one? 

The same sampling procedure occurred on each 33m transect line. Three transect lines were chosen to capture the range of variability in these areas. To account for the 3 transects at each edge site (note: two edge sites per location -St. Denis and Conservation Learning Centre), transects were nested within site and considered a random factor during our analyses. A supporting explanation of the sampling design is provided in the figure legend. We have also revised our terminology throughout to improve the clarity of our sampling design.

Comment 7:

Figure 2, why there are no values in the structural equation models? All are positive? 

This was a conceptual structural model; thus it has no values. The actual SEM models with values can be found in the last figure (now figure 4).

Comment 8:

I suggest combine Figure 3 and 6 together? Figure 4 and 5 into one plate, if not, could one of them putting in the appendix? Seven figures are too many I think. Figure 7 is fine, might put it as appendix for Figure 2? 

We thank the reviewer for their suggestion and agree seven figures was too many. In examining our figures again, we have focused the paper on four figures and placed the others (2,4,5) into the supplementary document. 

Reviewer 2

Comment 1:

This manuscript investigated the plant and soil communities along the grassland-cropland edge, generally the experiment was well designed, however, I am confused about the analysis about different sites, for the organizing of the results and figures. As you selected two sites to represent the edges, shall they be considered as replications? why you analyze them individually? The three transect in each site was not real replicate. So, I think the authors should considered carefully about their experiment design and analyze the data properly.

We appreciate the reviewer’s comments and provide clarification on our design, analyses, and reasoning here. In our mixed linear model used to analyze biomass and all soil parameters, location (i.e., the St. Denis and Conservation Learning Centre) and site (i.e. two edge sites sampled per location) and their interaction were included as fixed factors. Therefore, for these measurements all data were analyzed together. Furthermore, at each site within a given location, three transects were sampled for a total of six transects per location. Due to our design the transects were nested within site and included as a random factor in the mixed model. We have revised the terms used throughout the methods section to better clarify the design and explain the mixed linear model in the statistics section. 

NMDS analyses for the plant, bacteria and fungal communities, were originally analyzed including both locations in one analysis. However, due to strong differences in the plant community and management regimes between the two locations we found that analyzing each site separately provided more information regarding changes across the edge, which is the focus of our paper. The separation of the locations for the NMDS analyses was also supported by the differences in soil parameters found between the two locations. 

Comment 2:

The figures should be improved, especially for Figure 1 and 2, could be combined as one composite figure.

We have placed Figure 2 into the supplementary document and Figure 1 remains in the main document, as per reviewer 1’s suggestion.

Comment 3:

Another concerns is about the soil parameters, why you only select soil C and N to link with the microbial communities? have you think about the C/N ratios?

Your comment is greatly appreciated as soil C:N ratios are important when considering microbial communities. Initial exploratory analyses did not reveal strong trends with C:N ratios and we decided to include C and N only. We have added a point in the discussion regarding this topic to highlight the potential importance of C:N ratios.

---

## [Editor Report · Decision Letter 1]

20 Mar 2023

Agroecosystem edge effects on vegetation, soil properties, and the soil microbial community in the Canadian prairie

PONE-D-22-22877R1

Dear Dr. Stewart,

We’re pleased to inform you that your manuscript has been judged scientifically suitable for publication and will be formally accepted for publication once it meets all outstanding technical requirements.

Kind regards,

RunGuo Zang

Academic Editor

PLOS ONE

Additional Editor Comments (optional):

accept
---

## [Editor Report · Acceptance letter]

28 Mar 2023

PONE-D-22-22877R1 

Agroecosystem edge effects on vegetation, soil properties, and the soil microbial community in the Canadian prairie 

Dear Dr. Stewart:

I'm pleased to inform you that your manuscript has been deemed suitable for publication in PLOS ONE. Congratulations! Your manuscript is now with our production department. 

Kind regards, 

on behalf of

Professor RunGuo Zang 

Academic Editor

PLOS ONE